# Interpretability in the Era of Large Language Models: Opportunities and Challenges

## Abstract

Interpretable machine learning has exploded as an area of interest over the last decade, sparked by the rise of increasingly large datasets and deep neural networks. Simultaneously, large language models (LLMs) have demonstrated remarkable capabilities across a wide array of tasks, offering a chance to rethink opportunities in interpretable machine learning. Notably, the capability to explain in natural language allows LLMs to expand the scale and complexity of patterns that can be given to a human. However, these new capabilities raise new challenges, such as hallucinated explanations and immense computational costs.

In this position paper, we start by reviewing existing methods to evaluate the emerging field of LLM interpretation (both interpreting LLMs and using LLMs for explanation). We contend that, despite their limitations, LLMs hold the opportunity to redefine interpretability with a more ambitious scope across many applications, including in auditing LLMs themselves. We highlight two emerging research priorities for LLM interpretation: using LLMs to directly analyze new datasets and to generate interactive explanations.

## 1 Introduction

Machine learning (ML) and natural language processing (NLP) have seen a rapid expansion in recent years, due to the availability of increasingly large datasets and powerful neural network models. In response, the field of interpretable ML* has grown to incorporate a diverse array of techniques and methods for understanding these models and datasets (Doshi-Velez & Kim, 2017; Murdoch et al., 2019; Molnar, 2019). One part of this expansion has focused on the development and use of inherently interpretable models (Rudin et al., 2021), such as sparse linear models, generalized additive models, and decision trees. Alongside these models, post-hoc interpretability techniques have become increasingly prominent, offering insights into predictions after a model has been trained. Notable examples include methods for assessing feature importance (Ribeiro et al., 2016; Lundberg & Lee, 2017), and broader post-hoc techniques, e.g., model visualizations (Yosinski et al., 2015; Bau et al., 2018), or interpretable distillation (Tan et al., 2018; Ha et al., 2021).

Meanwhile, pre-trained large language models (LLMs) have shown impressive proficiency in a range of complex NLP tasks, significantly advancing the field and opening new frontiers for applications (Brown et al., 2020; Touvron et al., 2023; OpenAI, 2023). However, the inability to effectively interpret these models has debilitated their use in high-stakes applications such as medicine, and raised issues related to regulatory pressure, safety, and alignment (Goodman & Flaxman, 2016; Amodei et al., 2016; Gabriel, 2020). Moreover, this lack of interpretability has limited the use of LLMs (and other neural-network models) in fields such as science and data analysis (Wang et al., 2023a; Kasneci et al., 2023; Ziems et al., 2023). In these settings, the end goal is often to elicit a trustworthy interpretation rather than to deploy an LLM. In other settings, interpretability may instead be used as a tool to audit, understand, or manipulate LLMs.

In this work, we contend that LLMs hold the opportunity to rethink interpretability with a more ambitious scope. LLMs can elicit more elaborate explanations than the previous generation of interpretable ML techniques. While previous methods have often relied on restricted interfaces such as saliency maps, LLMs

---

*We use the terms interpretable, explainable, and transparent interchangeably.

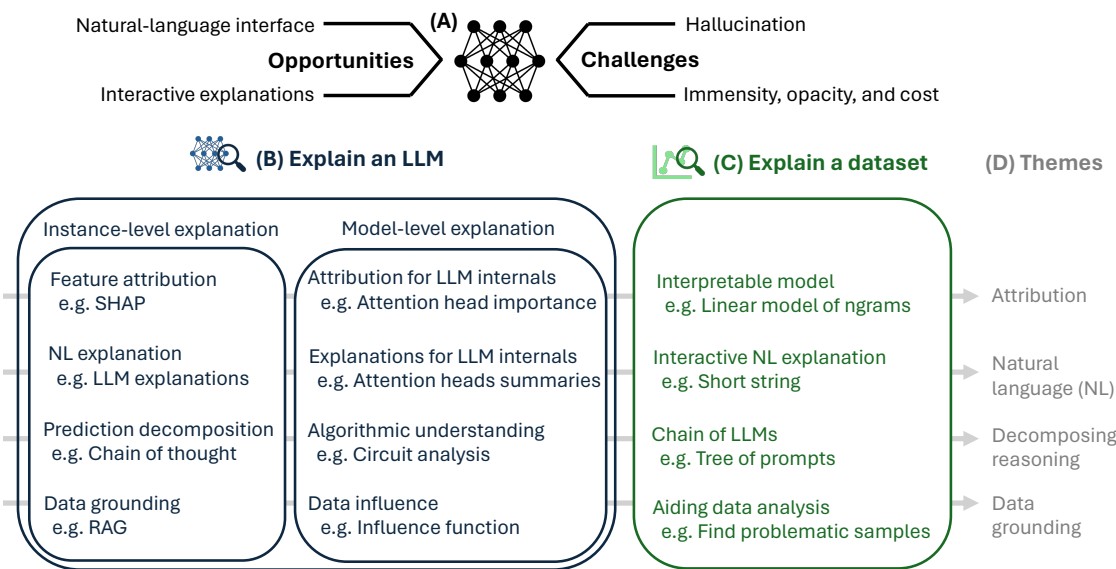

**Figure 1: Categorization of LLM interpretation research**. **(A)** LLMs raise unique opportunities and challenges for interpretation (Sec. 3). **(B)** Explaining an LLM can be categorized into methods that seek to explain a single generation from an LLM (i.e. instance-level explanation, Sec. 4.1) or the LLM in its entirety (i.e. model-level explanation, Sec. 4.2). Instance-level explanation methods build on many techniques that were originally developed for interpreting non-LLM models, such as feature attribution methods. More recent instance-level explanation techniques use LLMs themselves to yield interpretations, e.g., through post-hoc natural language (NL) explanations, asking an LLM to build explanations into its generation process, or through data grounding. Similar techniques have been developed and applied to model-level explanation, although it also includes unique types of explanations, e.g., analyzing individual attention heads or circuits inside an LLM. **(C)** Sec. 5 analyzes the emerging area that uses an LLM to aid in directly explaining a *dataset.* In this setting, an LLM is given a new dataset (which can consist of either text or tabular features) and is used to help analyze it. LLM-based techniques for dataset explanation are quite diverse, including helping to build interpretable models, generate NL explanations, generate chains of NL explanations, or construct data visualizations. **(D)** Common themes emerge among methods for instance-level explanation, model-level explanation, and dataset explanation.

can communicate directly in expressive natural language. This allows users to make targeted queries, such as *Can you explain your logic?*, *Why didn't you answer with (A)?*, or *Explain this data to me.*, and get immediate, relevant responses. We believe simple questions such as these, coupled with techniques for grounding and processing data, will allow LLMs to articulate previously incomprehensible model behaviors and data patterns directly to humans in understandable text. However, unlocking these opportunities requires tackling new challenges, including hallucinated (i.e. incorrect or baseless) explanations, along with the immense size, cost, and inherent opaqueness of modern LLMs.

**Contributions and overview**   We evaluate LLM interpretation and highlight emerging research priorities, taking a broader scope than recent works, e.g., those focused on explaining LLM predictions (Zhao et al., 2023), mechanistic interpretability (Räuker et al., 2023), social science (Ziems et al., 2023), or science more generally (Wang et al., 2023a; Birhane et al., 2023; Pion-Tonachini et al., 2021). Rather than providing an exhaustive overview of methods, we highlight the aspects of interpretability that are unique to LLMs and showcase them with practically useful methods.

Specifically, we begin with a background and definitions (Sec. 2) before proceeding to analyze the unique opportunities and challenges that LLMs present for interpretation (Sec. 3). We then ground these oppor-

tunities in two complementary categories for LLM-based interpretation (see Fig. 1). The first is generating explanations for *an existing LLM* (Sec. 4), which is useful for auditing a model's performance, alignment, fairness, etc. The second is explaining *a dataset* (Sec. 5); in this setting, an LLM is used to help analyze a new dataset (which can consist of either text or tabular features). While we cover these areas separately, there is a great deal of interplay between them. For example, explaining a dataset relies on many tools originally developed to extract reliable explanations when explaining an LLM. Conversely, explaining an LLM often requires understanding the dataset on which it was trained, as well as what dataset characteristics elicit different LLM behaviors.

Throughout the paper, we highlight dataset explanation and interactive explanation as emerging research priorities. Together, these two areas have great potential real-world significance in domains from science to statistics, where they can facilitate the process of scientific discovery, data analysis, and model building. Throughout, we focus on pre-trained LLMs, mostly applied to text data, but also applied to tabular data.

## 2 Background: definitions and evaluation

### 2.1 Definitions

**Interpretability**, when used without context, is a poorly defined concept. Precisely defining interpretability requires understanding the problem and audience an interpretation is intended to serve. In light of this imprecision, interpretable ML has largely become associated with a narrow set of techniques, including feature attribution, saliency maps, and transparent models. However, LLM interpretation is broader in scope and more expressive than these methods. Here, we paraphrase the definition of interpretable ML from Murdoch et al. 2019 to define LLM interpretation as the *extraction of relevant knowledge from an LLM concerning relationships either contained in data or learned by the model*. We emphasize that this definition applies to both interpreting an LLM and to using an LLM to generate explanations. Moreover, the definition relies on the extraction of *relevant* knowledge, i.e., knowledge that is useful for a particular problem and audience. For example, in a code generation context, a relevant interpretation may help a user quickly integrate an LLM-generated code snippet. In contrast, a relevant interpretation in a medical diagnosis setting may inform a user whether or not a prediction is trustworthy.

**Large language model** is a term that is often used imprecisely. Here, we use it to refer to transformer-based neural language models that contain tens to hundreds of billions of parameters, and which are pre-trained on massive text data, e.g., PaLM (Chowdhery et al., 2023), LLaMA (Touvron et al., 2023), and GPT-4 (OpenAI, 2023). Compared to early pre-trained language models (such as BERT (Devlin et al., 2018)), LLMs are not only much larger, but also exhibit stronger language understanding, generation abilities, and explanation capabilities. After an initial computationally intensive pre-training stage, LLMs often undergo instruction finetuning and further alignment with human preferences to improve instruction following (Ouyang et al., 2022) or to improve interactive chat capabilities, e.g., the LLaMA-2 chat model (Touvron et al., 2023). They are sometimes also further adapted via supervised finetuning to improve performance in a specific domain, such as medicine (Singhal et al., 2023).

**Prompting** is the most common interface for applying LLMs (and our main focus in this paper), once they have been trained. In prompting, a text prompt is fed into an LLM and used to generate subsequent output text, e.g. an answer to a question. *Few-shot prompting* is a type of prompting that involves providing an LLM with a small number of examples to allow it to better understand the task it is being asked to perform.

Interpretability is intricately related to other research areas where LLMs have begun to play an expanding role. For example, in the field of causal inference, interpreting and querying LLMs may help extract and test hypotheses for causal relationships contained in data (Kıcıman et al., 2023). Additionally, interpretability is a major topic when considering the bias, fairness, privacy, and security of LLMs (Yao et al., 2024; Li et al., 2023d), where it can be used to expose, fix, or exploit issues with an LLM.

## 2.2 Evaluating LLM interpretations

Evaluating interpretability is difficult in general (Doshi-Velez & Kim, 2017), requiring that interpretations be faithful to an underlying process in an LLM/dataset while remaining understandable to a human. This process is made even more difficult by the use of LLMs, which broadens the space of explanations and complicates rigorous understanding. In this section, we briefly overview three key approaches to evaluating LLM interpretations through (i) human studies, (ii) automated metrics, and (iii) using the interpretation to improve model performance. Careful consideration is required to match the choice of metric to the context of a particular problem.

**Evaluation with human studies**  Since different interpretations are relevant to different contexts, the ideal way to evaluate an interpretation is by studying whether its usage in a real-world setting with humans improves a desired outcome (Kim et al., 2017). This approach directly measures the utility of an interpretation in its appropriate context. However, these targeted human studies have some shortcomings: They are often laborious to conduct and are context-specific, i.e. an interpretation that helps humans in some contexts may not help them in others. Additionally, human studies that simply measure a proxy for usefulness (i.e. measuring human judgment of explanations) are often uninformative, as they may not translate into improvements in practice (Adebayo et al., 2018). A recent meta-analysis finds that introducing NLP explanations into settings with humans yields widely varying utilities, ranging from completely unhelpful to very useful (Chaleshtori et al., 2023). An important piece of this evaluation is the notion of complementarity (Bansal et al., 2021), i.e., that explanations should help LLMs complement human performance in a team setting, rather than improve their performance in isolation.

**Evaluation with automated metrics**  While human studies provide the most realistic evaluation, automated metrics (that can be computed without involving humans) are desirable to ease and scale evaluation, especially in model-level explanation. An increasingly popular approach is to use LLMs themselves in evaluation, although great care must be taken to avoid introducing biases, e.g., an LLM systematically scoring its own outputs too positively (Zheng et al., 2023). One way to reduce bias is to use LLMs as part of a structured evaluation process tailored to a particular problem, rather than directly querying LLMs for evaluation scores. For example, one common setting is evaluating a natural-language interpretation of a given function (which may be any component of a pre-trained LLM). In this setting, one can evaluate an explanation's ability to simulate the function's behavior (Bills et al., 2023), the function's output on LLM-generated synthetic data (Singh et al., 2023b), or its ability to recover a groundtruth function (Schwettmann et al., 2023; Zhong et al., 2023). In a question-answering setting, many automated metrics have been proposed for measuring the faithfulness of a natural-language explanation for an individual answer to a question (Atanasova et al., 2023; Parcalabescu & Frank, 2023; Chen et al., 2022).

**Evaluation through improving model performance**  A third avenue for evaluating interpretations is through their ability to control or improve model performance. This approach provides strong evidence for the utility of an explanation, although it does not encompass all critical use cases of interpretability (particularly those directly involving human interaction). Model improvements can take various forms, the simplest of which is simply improving accuracy at downstream tasks. For example, few-shot accuracy was seen to improve when aligning an LLM's rationales with explanations generated using post-hoc explanation methods (Krishna et al., 2023) or explanations distilled from large models (Mukherjee et al., 2023). Moreover, employing few-shot explanations during inference (not training) can significantly improve few-shot LLM accuracy, especially when these explanations are further optimized (Lampinen et al., 2022; Ye & Durrett, 2023). Beyond general performance, explanations can be used to overcome specific shortcomings of a model. For example, one line of work identifies and addresses shortcuts/spurious correlations learned by an LLM (Du et al., 2023; Kang & Choi, 2023; Bastings et al., 2021). Model editing, a related line of work, enables precise modifications to certain model behaviors, enhancing overall performance (Meng et al., 2022; Mitchell et al., 2022; Hernandez et al., 2023).

# 3 Unique opportunities and challenges of LLM interpretation

In this section, we discuss two unique opportunities and two challenges that arise in LLM interpretation (see Fig. 1A).

**Unique opportunities of LLM interpretation** First among LLM interpretation opportunities is the ability to provide *a natural-language interface* to explain complex patterns. This interface is familiar to humans, potentially ameliorating the difficulties that practitioners often face when using explainability techniques (Kaur et al., 2020; Weld & Bansal, 2019). Additionally, natural language can be used to build a bridge between humans and a range of other modalities, e.g., DNA, chemical compounds, or images (Taylor et al., 2022; Liu et al., 2023b; Radford et al., 2021), that may be difficult for humans to interpret on their own. In these cases, natural language allows for expressing complex concepts through explanations at different levels of granularity, potentially grounded in evidence or discussions of counterfactuals. For example, a natural language explanation for an LLM's answer to a question may highlight the LLM's coarse reasoning if the LLM is prompted for a high-level explanation or instead to specific words in the input if the LLM is asked for a more fine-grained explanation.

A second major opportunity is the ability for LLMs to generate *interactive explanations*. Interactivity allows users to tailor explanations to their unique needs, e.g., by asking follow-up questions and performing analysis on related examples. Interviews with decision-makers, including physicians and policymakers, indicate that they strongly prefer interactive explanations, particularly in the form of natural-language dialogues (Lakkaraju et al., 2022). Interactivity further allows LLM explanations to be decomposed into many different LLM calls, each of which can be audited independently. This can be enabled in different ways, e.g., having a user repeatedly chat with an LLM using prompting, or providing a user a sequence of LLM calls and evidence to analyze.

**Unique challenges of LLM interpretation** These opportunities bring new challenges. First and foremost is the issue of *hallucination*, i.e. incorrect or baseless LLM generations. In our setting, we focus on hallucinated explanations generated by an LLM; Flexible explanations provided in natural language can quickly become less grounded in evidence, whether the evidence is present in a given input or presumed to be present in the knowledge an LLM has learned from its training data. Hallucinated explanations are unhelpful or even misleading, and thus techniques for identifying and combating hallucination are critical to the success of LLM interpretation.

A second challenge is the *immensity and opaqueness* of LLMs. Models have grown to contain tens or hundreds of billions of parameters (Brown et al., 2020; Touvron et al., 2023), and continue to grow in size. This makes it infeasible for a human to inspect or even comprehend the units of an LLM. Moreover, it necessitates efficient algorithms for interpretation, as even generating a single token from an LLM often incurs a non-trivial computational cost. In fact, LLMs are often too large to be run locally or can be accessed only through a proprietary text API, necessitating the need for interpretation algorithms that do not have full access to the model (e.g., no access to the model weights or the model gradients).

# 4 Explaining an LLM

In this section, we study techniques for explaining an LLM, including explaining a single generation from an LLM (Sec. 4.1) or an LLM in its entirety (Sec. 4.2). Most of the methods we cover are specific to LLMs, but we also cover some general interpretable ML techniques (e.g. feature attributions) as well as some methods that apply to neural networks that are not LLMs (e.g. probing); see a full breakdown in Table 1.

**Table 1:** Breakdown of LLM interpretation methods. First categorizes methods based on the model class they apply to: LLMs, deep neural networks (DNNs), or general ML methods (ML). Additionally categorizes methods based on whether they fall under instance-level interpretation (I-L), model-level interpretation (M-L), and whether they us an LLM as a tool to generate natural-language explanations (N-L). All methods in Sec. 5 use LLMs as a tool generate different forms of interpretations at the dataset-level.

| Methods | Example works | Applicability | Usage |
|---|---|---|---|
| Feature attributions | Lundberg & Lee 2017;Enguehard 2023 | ML | I-L |
| Interpreting attention scores | Jain & Wallace 2019;Bibal et al. 2022 | LLM | I-L |
| Natural-language explanations | Bhattacharjee et al. 2023;Chen et al. 2023b | LLM | I-L, N-L |
| Chain of thought | Wei et al. 2022 | LLM | I-L, N-L |
| RAG | Guu et al. 2020;Worledge et al. 2023 | LLM | I-L |
| Probing | Conneau et al. 2018;Zou et al. 2023 | DNN | M-L |
| Understanding internal model components | Mu & Andreas 2020;Singh et al. 2023b | DNN | M-L |
| Toy LLM models | Elhage et al. 2021;Olsson et al. 2022 | LLM | M-L, N-L |
| Investigating training data | Grosse et al. 2023;Kandpal et al. 2023 | ML | M-L |
| Chat-based interactivity | Slack et al. 2022;Wang et al. 2024 | LLM | M-L, N-L |

### 4.1 Instance-level explanation

Instance-level explanation, i.e., explaining a single generation from an LLM, has been a major focus in the recent interpretability literature. It allows for understanding and using LLMs in high-stakes scenarios, e.g., healthcare.

**Feature attributions** The simplest approach for providing instance-level explanations in LLMs provides feature attributions for input tokens. These feature attributions assign a relevance score to each input feature, reflecting its impact on the model's generated output. Various attribution methods have been developed, including perturbation-based methods (Lundberg & Lee, 2017), gradient-based methods (Sundararajan et al., 2017; Montavon et al., 2017; Li et al., 2015), and linear approximations (Ribeiro et al., 2016). Recently, these methods have been specifically adapted for transformer models, addressing unique challenges such as discrete token embeddings (Sikdar et al., 2021; Enguehard, 2023) and computational costs (Chen et al., 2023a). Moreover, the conditional distribution learned by an LLM can be used to enhance existing attribution methods, e.g., by performing input marginalization (Kim et al., 2020). These works focus on attributions at the token-level, but may consider using a less granular breakdown of the input to improve understandability (Zafar et al., 2021).

**Interpreting attention scores** Besides feature attributions, attention mechanisms within an LLM offer another avenue for visualizing token contributions to an LLM generation (Wiegreffe & Pinter, 2019), though their faithfulness/effectiveness remains unclear (Jain & Wallace, 2019; Bibal et al., 2022). Interestingly, recent work suggests that LLMs themselves can generate post-hoc attributions of important features through prompting (Kroeger et al., 2023). This approach could be extended to enable eliciting different feature attributions that are relevant in different contexts.

**Natural-language explanations** Beyond token-level attributions, LLMs can also generate instance-level explanations directly in natural language. While the generation of natural-language explanations predates the current era of LLMs (e.g., in text classification (Camburu et al., 2018; Rajani et al., 2019) or image classification (Hendricks et al., 2016)), the advent of more powerful models has significantly enhanced their effectiveness. Natural-language explanations generated by LLMs have shown the ability to elucidate model predictions, even simulating counterfactual scenarios (Bhattacharjee et al., 2023), and expressing nuances like uncertainty (Xiong et al., 2023; Tanneru et al., 2023; Zhou et al., 2024). Despite their potential benefits, natural language explanations remain extremely susceptible to hallucination or inaccuracies, especially when generated post-hoc (Chen et al., 2023b; Ye & Durrett, 2022).

**Chain of thought** One starting point for combating these hallucinations is integrating an explanation within the answer-generation process itself. Chain-of-thought prompting exemplifies this approach (Wei et al., 2022), where an LLM is prompted to articulate its reasoning step-by-step before arriving at an answer. This reasoning chain generally results in more accurate and faithful outcomes, as the final answer is often more aligned with the preceding logical steps. The faithfulness of the produced step-by-step explanation can be tested by introducing perturbations in the reasoning process and observing the effects on the final output (Madaan & Yazdanbakhsh, 2022; Wang et al., 2022a; Lanham et al., 2023). Alternative methods for generating this reasoning chain exist, such as tree-of-thoughts (Yao et al., 2023), which extends chain-of-thought to instead generate a tree of thoughts used in conjunction with backtracking, graph-of-thoughts (Besta et al., 2023), and others (Nye et al., 2021; Press et al., 2022; Zhou et al., 2022). All of these methods not only help convey an LLM's intermediate reasoning to a user, but also help the LLM to follow the reasoning through prompting, often enhancing the reliability of the output. However, like all LLM-based generations, the fidelity of these explanations can vary (Lanham et al., 2023; Wei et al., 2023).

**RAG** An alternative path to reducing hallucinations during generation is to employ retrieval-augmented generation (RAG). In RAG, an LLM incorporates a retrieval step in its decision-making process, usually by searching a reference corpus or knowledge base using text embeddings (Guu et al., 2020; Peng et al., 2023); see review (Worledge et al., 2023). This allows the information that is used to generate an output to be specified and examined explicitly, making it easier to explain the evidence an LLM uses during decision-making.

## 4.2 Model-level explanation

Rather than studying individual generations, model-level explanation aims to understand an LLM as a whole, usually by analyzing its parameters. Recently, many model-level explanations of how model parameters algorithmically yield model behaviors have been labeled *mechanistic interpretability*, though the use/omission of this label is often imprecise, e.g. see (Räuker et al., 2023). Model-level explanations can help to audit a model for concerns beyond generalization, e.g., bias, privacy, and safety, helping to build LLMs that are more efficient / trustworthy. They can also yield mechanistic understanding about how LLMs function. Towards this end, researchers have focused on summarizing the behaviors and inner workings of LLMs through various lenses. Generally, these works require access to model weights and do not work for explaining models that are only accessible through a text API, e.g., GPT-4 (OpenAI, 2023).

**Probing** One popular method for understanding neural-network representations is probing. Probing techniques analyze a model's representation by decoding embedded information (Adi et al., 2016; Conneau et al., 2018; Giulianelli et al., 2018). In the context of LLMs, probing has evolved to include the analysis of attention heads (Clark et al., 2019), embeddings (Morris et al., 2023a), and different controllable aspects of representations (Zou et al., 2023). More recent methods directly decode an output token to understand what is represented at different positions and layers (Belrose et al., 2023; Ghandeharioun et al., 2024) or connecting multimodal embeddings with text embeddings to help make them more understandable (Oikarinen & Weng, 2022; Oikarinen et al., 2023; Bhalla et al., 2024). These methods can provide a deeper understanding of the nuanced ways in which LLMs process and represent information.

**Understanding internal model components** In addition to probing, many works study LLM representations at a more granular level. This includes categorizing or decoding concepts from individual neurons (Mu & Andreas, 2020; Gurnee et al., 2023; Dalvi et al., 2019; Lakretz et al., 2019) or directly explaining the function of attention heads in natural language (Bills et al., 2023; Hernandez et al., 2022b). Beyond individual neurons, there is growing interest in understanding how groups of neurons combine to perform specific tasks, e.g., finding a circuit for indirect object identification (Wang et al., 2022b), for entity binding (Feng & Steinhardt, 2023), or for multiple shared purposes (Merullo et al., 2023). More broadly, this type of analysis can be applied to localize functionalities rather than fully explain a circuit, e.g., localizing factual knowledge within an LLM (Meng et al., 2022; Dai et al., 2021). A persistent problem with these methods is that they are difficult to scale to immense LLMs, leading to research in (semi)-automated methods that can scale to today's largest LLMs (Lieberum et al., 2023; Wu et al., 2023).

**Toy LLM models** A complementary approach to mechanistic understanding uses miniature LLMs as a test bed for investigating complex phenomena. For example, examining a 2-layer transformer model reveals

information about what patterns are learned by attention heads as a function of input statistics (Elhage et al., 2021) or helps identify key components, such as induction heads or ngram heads that copy and utilize relevant tokens (Olsson et al., 2022; Akyürek et al., 2024). This line of mechanistic understanding places a particular focus on studying the important capability of in-context learning, i.e., given a few input-output examples in a prompt, an LLM can learn to correctly generate an output for a new input (Garg et al., 2022; Zhou et al., 2023).

**Investigating training data**  A related area of research seeks to interpret an LLM by understanding the influence of its training data distribution. Unlike other methods we have discussed, this requires access to an LLM's training dataset, which is often unknown or inaccessible. In the case that the data is known, researchers can employ techniques such as influence functions to identify important elements in the training data (Grosse et al., 2023). They can also study how model behaviors arise from patterns in training data, such as hallucination in the presence of long-tail data (Kandpal et al., 2023), in the presence of repeated training data (Hernandez et al., 2022a), in-context learning (Chen et al., 2024b), statistical patterns that contradict proper reasoning (McKenna et al., 2023), and others (Swayamdipta et al., 2020).

**Chat-based interactivity**  All these interpretation techniques can be improved via LLM-based interactivity, allowing a user to investigate different model components via follow-up queries and altered prompts from a user. For example, one recent work introduces an end-to-end framework for explanation-based debugging and improvement of text models, showing that it can quickly yield improvements in text-classification performance (Lee et al., 2022). Another work, Talk2Model, introduces a natural-language interface that allows users to interrogate a tabular prediction model through a dialog, implicitly calling many different model explainability tools, such as calculating feature importance (Slack et al., 2022).[†] More recent work extends Talk2Model to a setting interrogating an LLM about its behavior (Wang et al., 2024).

**Model manipulation**  Finally, the insights gained from model-level understanding are beginning to inform practical applications, with current areas of focus including model editing (Meng et al., 2022), improving instruction following (Zhang et al., 2023b), and model compression (Sharma et al., 2023). These areas simultaneously serve as a sanity check on many model-level interpretations and as a useful path to enhancing the reliability of LLMs.

# 5 Explaining a dataset

As LLMs improve their context length and capabilities, they can be leveraged to explain an entire dataset, rather than explaining an LLM or its generations. This can aid with data analysis, knowledge discovery, and scientific applications. Fig. 2 shows an overview of dataset explanations at different levels of granularity, which we cover in detail below. We distinguish between tabular and text data, but note that most methods can be successfully applied to either, or both simultaneously in a multimodal setting.

## 5.1 Tabular dataset explanation

**LLM-aided data analysis**  One way LLMs can aid in dataset explanation is by making it easier to interactively visualize and analyze tabular data. This is made possible by the fact that LLMs can simultaneously understand code, text, and numbers by treating them all as input tokens. Perhaps the most popular method in this category is ChatGPT Code Interpreter[‡], which enables uploading datasets and building visualizations on top of them through an interactive text interface. Underlying this interface, the LLM makes calls to tools such as python functions to create the visualizations. This capability is part of a broader trend of LLM-aided visualization, e.g., suggesting automatic visualizations for dataframes (Dibia, 2023), helping to automate data wrangling (Narayan et al., 2022), or even conducting full-fledged data analysis (Huang et al., 2023a) with accompanying write-ups (Ifargan et al., 2024). These capabilities benefit from a growing line of work that analyzes how to effectively represent and process tabular data with LLMs (Li et al., 2023b; Zhang et al., 2023a;c).

---

[†]Note that Talk2Model focuses on interpreting prediction models rather than LLMs.
[‡]https://openai.com/blog/chatgpt-plugins#code-interpreter

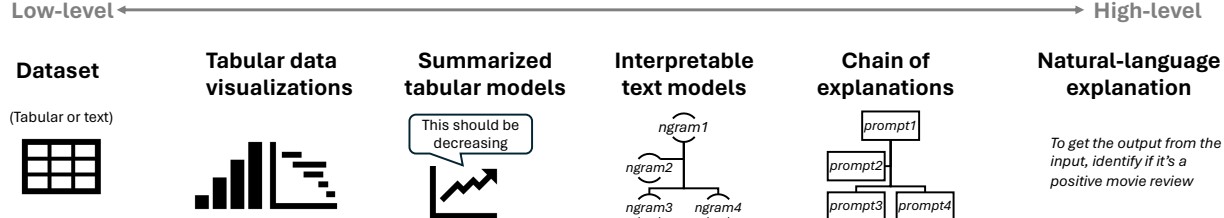

**Figure 2: Dataset explanations at different levels of granularity.** Dataset explanation involves understanding a new dataset (consisting of either text or tabular features) using a pre-trained LLM. Low-level explanations are more faithful to the dataset but involve more human effort to extract meaningful insights. Many dataset interpretations use prediction models (classification or regression) as a means to identify and explain patterns between features.

**Explaining models fit to tabular datasets** LLMs can also help explaining datasets by directly analyzing models that have been fit to tabular data. Unlike model-level explanation, where the goal is to understand the model, in dataset explanation, the goal is to understand patterns in the data through the model (although similar techniques can be used for both problems). For example, one recent work uses LLMs to analyze generalized additive models (GAMs) that are fit to tabular data (Lengerich et al., 2023). GAMs are interpretable models that can be represented as a set of curves, each representing the contribution of a feature to the output prediction as a function of the feature's value. An LLM can analyze the fitted model (and thereby the underlying dataset) by processing each curve as a set of numerical tokens and then detecting and describing patterns in each curve. Lengerich et al. find that LLMs can identify surprising characteristics in the curves and the underlying data, largely based on their prior knowledge of a domain. Rather than using an interpretable GAM model, another approach is to distill dataset insights by analyzing classifier predictions. For example, MaNtLE generates natural-language descriptions of a classifier's rationale based on the classifier's predictions, and these explanations are found to identify explainable subgroups that contain similar feature patterns (Menon et al., 2023).

## 5.2 Text dataset explanation

**Fully interpretable models** Text data poses different challenges for dataset explanation than tabular data because it is sparse, high-dimensional, and modeling it requires many high-order interactions. As a result, interpretable models that have been successful in the tabular domain (e.g., sparse linear models (Tibshirani, 1996; Ustun & Rudin, 2016), GAMs (Hastie & Tibshirani, 1986; Lou et al., 2013; Caruana et al., 2015), decision trees (Breiman et al., 1984; Quinlan, 1986; Agarwal et al., 2022), and others (Rudin, 2018)), have struggled to accurately model text. One recent line of work addresses this issue by using LLMs to help build fully interpretable text models, such as linear models or decision trees (Singh et al., 2023a); the resulting models are surprisingly accurate, often outperforming even much larger LLM models. These interpretable models can help explain a dataset by showing which features (i.e. words or ngrams) are important for predicting different outcomes. Similar methods, use LLMs to build interpretable representations for text classification (McInerney et al., 2023), text style (Patel et al., 2023), or text regression (Benara et al., 2024).

**Partially interpretable models** Going beyond fully interpretable models, LLMs also help in building partially interpretable text models. Partially interpretable text models often employ chains of prompts; these chains allow for decomposing an LLM's decision-making process to analyze which dataset patterns a model learns. Prompt chains are usually constructed by humans or by querying a model to generate a chain of calls on-the-fly (Grunde-McLaughlin et al., 2023). For dataset explanation, the most relevant chains are sequences of explanations that are generated by an LLM. For example, a model can generate a single tree of explanations that is shared across all examples in a dataset, a process that enables understanding hierarchical structures stored within a dataset (Morris et al., 2023b). Rather than a tree, a single chain of prompts can often help an LLM employ self-verification, i.e. the model itself checks its previous generations

using a chain of prompts, a popular technique that often improves reliability (Pan et al., 2023; Madaan et al., 2023; Gero et al., 2023). As in instance-level explanation, an LLM can incorporate a retrieval step in its decision-making process (Worledge et al., 2023), and access to different tools can help make different steps (e.g., arithmetic) more reliable and transparent (Mialon et al., 2023).

**Natural-language explanations**  Natural-language explanations hold the potential to produce rich, concise descriptions of patterns present in a dataset, but are prone to hallucination. One method, iPrompt (Singh et al., 2023c), aims to avoid hallucination by searching for a dataset explanation in the form of a single prompt, and verifying that the prompt induces an LLM to accurately predict a pattern in the underlying dataset. Another work learns and verifies a library of natural-language rules that help improve question answering (Zhu et al., 2023). Related methods use LLMs to provide descriptions that differentiate between groups in a dataset, followed by an LLM that verifies the credibility of the description (Zhong et al., 2022; 2023; Zhu et al., 2022). In addition to a raw natural-language explanation, LLMs can aid in summarizing textual information, e.g., through explainable clustering of a text dataset (Wang et al., 2023b) or creating prompt-based topic models (Pham et al., 2023).

## 6  Future research priorities

We now highlight research priorities surrounding LLM interpretation in three areas: explanation reliability, knowledge discovery, and interactive explanations.

**Explanation reliability**  All LLM-generated explanations are bottlenecked by reliability issues. This includes hallucinations (Tonmoy et al., 2024), but encompasses a broader set of issues. For example, LLMs continue to be very sensitive to the nuances of prompt phrasing; minor variations in prompts can completely change the substance of an LLM output (Sclar et al., 2023; Turpin et al., 2023). Additionally, LLMs may ignore parts of their context, e.g., the middle of long contexts (Liu et al., 2023a) or instructions that are difficult to parse (Zhang et al., 2023b).

These reliability issues are particularly critical in interpretation, which often uses explanations to mitigate risk in high-stakes settings. LLM interpretations rarely come with theoretical guarantees of reliability, and those that do are often for very constrained settings. One work analyzing explanation reliability finds that LLMs often generate seemingly correct explanations that are actually inconsistent with their own outputs on related questions (Chen et al., 2023b), preventing a human practitioner from trusting an LLM or understanding how its explanations apply to new scenarios. Another study finds that explanations generated by an LLM may not entail the model's predictions or be factually grounded in the input, even on simple tasks with extractive explanations (Ye & Durrett, 2022).

Future work is required to improve the grounding of explanations and develop stronger, computationally tractable methods to test their reliability, perhaps through methods such as RAG (Worledge et al., 2023; Patel et al., 2024), self-verification (Pan et al., 2023), iterative prompting/grounding (Singh et al., 2023c), or automatically improving model self-consistency (Chen et al., 2024a; Li et al., 2023c; Akyürek et al., 2024).

**Knowledge discovery (through dataset explanation)**  Dataset explanation using LLMs (Sec. 5) holds the potential to help with the generation and discovery of new knowledge from data (Wang et al., 2023a; Birhane et al., 2023; Pion-Tonachini et al., 2021), rather than simply helping to speed up data analysis or visualization. Dataset explanation could initially help at the level of brainstorming scientific hypotheses that can then be screened or tested by human researchers (Yang et al., 2023). During and after this process, LLM explanations can help with using natural language to understand data from otherwise opaque domains, such as chemical compounds (Liu et al., 2023b) or DNA sequences (Taylor et al., 2022). In the algorithms domain, LLMs have been used to uncover new algorithms, translating them to humans as readable computer programs (Romera-Paredes et al., 2023). These approaches could be combined with data from experiments to help yield new data-driven insights.

LLM explanations can also be used to help humans better understand and perform a task. Explanations from transformers have already begun to be applied to domains such as Chess, where their explanations

can help improve even expert players (Schut et al., 2023). Additionally, LLMs can provide explanations of expert human behavior, e.g. "Why did the doctor prescribe this medication given this information about the patient?", that are helpful in understanding, auditing, and improving human behavior (Tu et al., 2024).

**Interactive explanations**  Finally, advancements in LLMs are poised to allow for the development of more user-centric, interactive explanations. LLM explanations and follow-up questions are already being integrated into a variety of LLM applications, such as interactive task specification (Li et al., 2023a), recommendation (Huang et al., 2023b), and a wide set of tasks involving dialog. Furthermore, works like Talk2Model (Slack et al., 2022) enable users to interactively audit models in a conversational manner. This dialog interface could be used in conjunction with many of the methods covered in this work to help with new applications, e.g., interactive dataset explanation.

## 7 Conclusions

In this paper, we have explored the vast and dynamic landscape of interpretable ML, particularly focusing on the unique opportunities and challenges presented by LLMs. LLMs' advanced natural language generation capabilities have opened new avenues for generating more elaborate and nuanced explanations, allowing for a deeper and more accessible understanding of complex patterns in data and model behaviors. As we navigate this terrain, we assert that the integration of LLMs into interpretative processes is not merely an enhancement of existing methodologies but a transformative shift that promises to redefine the boundaries of machine learning interpretability.

Our position is anchored in the belief that the future of interpretable ML hinges on our ability to harness the full potential of LLMs. To this end, we outlined several key stances and directions for future research, such as enhancing explanation reliability and advancing dataset interpretation for knowledge discovery. As LLMs continue to improve rapidly, these explanations (and all the methods discussed in this work) will advance correspondingly to enable new applications and insights. In the near future, LLMs may be able to offer the holy grail of interpretability: explanations that can reliably aggregate and convey extremely complex information to us all.

## Broader impact statement

This paper presents work whose goal is to advance the field of LLM interpretation, a crucial step toward addressing the challenges posed by these often opaque models. Although LLMs have gained widespread use, their lack of transparency can lead to significant harm, underscoring the importance of interpretable AI. There are many potential positive societal consequences of this form of interpretability, e.g., facilitating a better understanding of LLMs and how to use them safely, along with a better understanding of scientific data and models. Nevertheless, as is the case with most ML research, the interpretations could be used to interpret and potentially improve an LLM or dataset that is being used for nefarious purposes.

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
