# OpenReview forum: "Interpretability in the Era of Large Language Models: Opportunities and Challenges"
_TMLR — Rejected by TMLR_

### Review · Reviewer_hoaw · 2024-06-18

**Summary Of Contributions:**

This paper offers an overview of the literature pertaining to LLMs and their use for and within interpretability methods. From this review, the authors argue that two priorities emerge, research on LLMs-as-dataset-interpreters, and LLMs-as-interactive-explainers.

**Audience:**

Yes

**Broader Impact Concerns:**

No concerns.

**Claims And Evidence:**

No

**Requested Changes:**

I think the two above questions being answered within the paper would make for a stronger work.

Regarding the second question, an easy stance on interpretability, is that it feels like solving the proposed problems would amount to just solving underlying problems (i.e. making better architectures/learning methods). Why not solve them in the first place?

I would like to see it argued in the paper that interpretability is a “worthwhile” endeavour. The authors state that interpretability research is not “merely an enhancement” of existing methods. I would tend to agree, but this is not argued in depth in the paper. The alternative is that existing methods, deep learning, transformers, maximum likelihood models, are simply not compatible with our desire to “truly understand” the predictions of our models, and that novel methods are required (e.g. methods that perform causal reasoning explicitly).

Some nitpicking:
There’s this claim that CoT helps “convey an LLM’s intermediate reasoning”, yes and no, this is still truthy next token prediction. This is not _necessarily_ the LLM’s reasoning, it’s the LLM creating a thruthy output that also contains a truthy sounding explanation. Yes, in practice CoT increases the accuracy of predictions, but there’s nothing pushing the model to perform logical reasoning like A & B therefore C, only p(A, B, C). It just so happens that p(explanation, output) is marginally more likely to produce accurate outputs, possibly because the joint biases the marginal over outputs.

**Strengths And Weaknesses:**

Overall the paper does a decent job of surveying the field. It is illustrative and tries to be comprehensive. As a review paper, it connects and contextualizes the different approaches and research paths.

For an opinion paper, this work doesn’t seem to be very opinionated. For example, two aspects should be much clearer:
- which of the authors’ beliefs go counter to general opinion? Why?
- why should we care about interpretability over methodological improvement? and specifically, why the two championed directions of research? What are the alternatives and why not those?

Rating this paper is complicated by the lack of empirical results. Survey/opinion papers are perfectly fine for TMLR, and it’s not that there’s no contributions here, but those are informational rather than technical; evaluating their usefulness seems even more subjective than for the latter. Thus, the lack of very clear stances make me more uncertain in my evaluation.

---

> ### Author Response · Authors · 2024-07-08
> **Thank you for your review**
>
> We sincerely thank the reviewer for their thoughtful comments and time. We have made several changes to the paper in the revised version to incorporate the feedback, resulting in a much improved paper.
>
> (1) **Re: “which of the authors’ beliefs go counter to general opinion? Why?”** – The paper’s position itself is fairly simple: that LLMs will redefine interpretability with a more ambitious scope and that 2 emerging priorities here are in analyzing new datasets and generating interactive explanations. This stance may seem tame to LLM practitioners, but remains controversial in many disciplines including many of the folks that may benefit from LLM interpretability: e.g. scientists, medical practitioners, social scientists, and statisticians. We believe this work will help spur work in developing interpretability tools that can be used in applications today, and improving interpretability that can soon be used in new scenarios such as knowledge discovery or high-stakes decision making.
>
> Amidst this general stance, the paper’s focus on dataset explanation, interactive explanation, and natural-language explanations seem to be the pieces that are missing from the current literature on LLM interpretability. We note that a core contribution of this work itself is simply organizing the rapidly growing literature on LLM interpretability (e.g. 90 of the papers we reference are published just since the start of last year).
>
> (2) **Re: “why should we care about interpretability over methodological improvement?** – This is an important question and we have tried to clarify, e.g. with this addition to our 2nd paragraph of the introduction: “... the inability to effectively interpret these models has debilitated their use in high-stakes applications such as medicine, and raised issues related to regulatory pressure, safety, and alignment (Goodman & Flaxman, 2016; Amodei et al., 2016; Gabriel, 2020). Moreover, this lack of interpretability has limited the use of LLMs (and other neural-network models) in fields such as science and data analysis (Wang et al., 2023a; Kasneci et al., 2023; Ziems et al., 2023). *In these settings, the end goal is often to elicit a trustworthy interpretation rather than to deploy an LLM. In other settings, interpretability may instead be used as a tool to audit, understand, or manipulate LLMs.*”
>
> Generally, we do not see a dichotomy between interpretability and methodological improvement. For example, Section 6 discusses 2 future research priorities (dataset explanation and interactive explanations), both of which hinge on the reliable application of methodological developments in LLMs. Thus, many current challenges in interpretability research are specific forms of methodological improvement, e.g. how to provide grounding for natural-language explanations.
>
> (3) **Re: “why the two championed directions of research? What are the alternatives and why not those?”** – We highlight these two championed directions of research (dataset explanations and interactive explanations) as they are real-world applications for LLM interpretability that (1) can have very high impact, (2) are unique to LLMs, rather than neural networks/machine-learning models in general, and (3) are nascent / relatively unexplored. One alternative approach is mechanistic interpretability. Compared to our two directions, we feel mechanistic interpretability is already relatively popular as a research area [1], and thus we instead include a discussion of it when covering related work in Section 4. Other alternatives are often too specific for a review of this scope (e.g. LLMs for medicine) or are too general to be interpretability-focused (e.g. reducing hallucinations).
>
> (4) **Re: CoT faithfulness** – Agreed, a CoT explanation is not faithful in the same sense that a causal explanation can be. We add this note to qualify CoT explanations in the paper: “This reasoning chain generally results in more accurate and faithful outcomes, as the final answer is often more aligned with the preceding logical steps. The faithfulness of the produced step-by-step explanation can be tested by introducing perturbations in the reasoning process and observing the effects on the final output (Madaan & Yazdanbakhsh, 2022; Wang et al., 2022a; Lanham et al., 2023)”
>
> [1] https://arxiv.org/abs/2207.13243

---

### Review · Reviewer_MMFT · 2024-06-22

**Summary Of Contributions:**

This position paper outlined several directions for LLM interpretation, including interpreting LLMs and leveraging LLMs for dataset explanation. For interpreting LLMs, the authors categorized the existing literature into instance-level and model-level explanations. Instance-level explanations included feature attributions and natural-language explanations, while the hallucinations or inaccuracies of the latter’s explanations could be mitigated by integrating interpretation in the answer generation process (e.g., chain-of-thought) or retrieval-enhanced generation. Model-level explanations typically required access to model weights, such as probing and neuron-level explanations. For explaining a dataset, LLMs served as language/visualization tools to explain and analyze entire datasets, including both tabular data and text data.

**Audience:**

Yes

**Broader Impact Concerns:**

The paper should futher discuss the faithfulness of interpretation methods.

**Claims And Evidence:**

Yes

**Requested Changes:**

The authors should provide clear statements concerning the objectives, strengths, limitations, and relevant evaluation metrics (if available) for various types of interpretation methods. In addition, a distinct section should be dedicated to exploring the fidelity of these methods.  This section should discuss whether there are quantifiable metrics for evaluating fidelity, the theoretical guarantees of interpretation methods, and the extent to which interpretation results can be considered trustworthy.

To enhance this work, the authors may clarify the differences between traditional interpretable methods and those unique to LLMs.

**Strengths And Weaknesses:**

Strengths:

This is a timely position paper in which the authors summarize the opportunities and challenges in the field of interpretability during the era of LLMs, and outline potential directions for LLM interpretability.

Weaknesses:

1. The main weakness is the lack of a clear distinction between "interpreting LLMs," "understanding the mechanisms/behaviors of LLMs," and "leveraging LLMs to generate natural language explanations", each of which pursues distinct objectives. The first category, within the domain of interpretability of deep learning models, focuses on whether the provided explanation faithfully reflects the model’s decision-making process. These methods can be evaluated based on their reliability, fidelity, and computational complexity.
The second category aims to comprehend why deep learning models possess remarkable representation capabilities and to identify their inherent flaws.
The third category involves utilizing LLMs as tools to elucidate downstream tasks, such as explaining datasets, identifying significant elements within datasets, or employing explanation-based debugging. Therefore, the authors need to clearly distinguish between interpretations with varying objectives and, if possible, highlight the objections, strengths, limitations, and evaluation metrics of each category of LLM interpretations.

2. The paper lacks a clear distinction between traditional interpretable methods and those unique to LLMs (e.g., in-context learning and chain-of-thought). The authors need to highlight the distinctions and challenges of interpreting LLMs compared to interpreting general DNNs (i.e., arbitrary black-box models) [1]. For example, the paper should demonstrate which interpretation methods are universally applicable to DNNs, which are suitable for smaller models, and which are exclusive to LLMs. Additionally, it needs to evaluate the fidelity of these methods, outlining their strengths, weaknesses, and evaluation metrics.

3. The paper lacks a discussion on evaluating the faithfulness of interpretation methods. Although different levels of interpreting methods are listed, it's crucial to assess the fidelity of these methods and determine whether humans can trust the results. The paper needs to discuss the evaluation metrics for faithfulness, specifying which methods are theoretically guaranteed and which are empirical. Besides, in Section 2, "Evaluating LLM interpretations," the authors suggest evaluating an interpretation based on its ability to improve model outcomes. However, while an interpretation might enhance performance in some tasks, it may not necessarily be suitable for others. Therefore, the paper should evaluate an interpretation method from multiple perspectives, including fidelity/reliability, potential applications, and limitations of the interpretation outcomes.

[1] Zhao et al. Explainability for Large Language Models: A Survey, 2023.

---

> ### Author Response · Authors · 2024-07-08
> **Thank you for your review**
>
> We sincerely thank the reviewer for their thoughtful comments and time. We have made several changes to the paper in the revised version to incorporate the feedback, resulting in a much improved paper.
>
> 1. **Re: distinguishing between distinct objectives of LLM interpretations** – Thank you for your careful thought on these issues. We have added a new Table 1 which clarifies which of these different categories each approach belongs to (mapping “interpreting LLMs” as instance-level interpretation, i.e. explaining a single generation by an LLM and “understanding the mechanisms/behaviors of an LLM” to model-level explanation). Leveraging LLMs to generate natural language explanations crops up in both of these scenarios (as well as dataset explanation) and hopefully the table helps make this clear.
> 2. **Re: distinguishing between interpretable methods and those unique to LLMs** – Thank you for raising this point. Indeed, we had initially structured our paper sections around differences between methods, but later shifted it to structure around differences between interpretability problems. Section 3 tries to clarify the unique opportunities/challenges of LLM interpretation, and we have made changes to Section 4 and Section 5 to distinguish between general interpretability methods and those unique to LLMs. The main addition is also in the new Table 1, where we add a column describing the applicability of different approaches (and add appropriate headers / discussion in the text).
> 3. **Re: explanation faithfulness** – We have moved evaluation to its own section (“2.2 Evaluating LLM Interpretations”) and added many notes on faithfulness throughout the manuscript:
>     - “Evaluating interpretability is difficult in general (Doshi-Velez & Kim, 2017), requiring that interpretations be faithful to the underlying process they describe while remaining understandable to a human. This process is made even more difficult by the use of LLMs, which broadens the space of explanations and complicates rigorous understanding. In this section, we briefly overview three key approaches to evaluating LLM interpretations through (i) human studies, (ii) automated metrics, and (iii) using the interpretation to improve model performance.”
>     - “Careful consideration is required to match the choice of metric to the context of a particular problem.”
>     - “However, these targeted human studies have some shortcomings: They are often laborious to conduct and are context-specific, i.e. an interpretation that helps humans in some contexts may not help them in others. Additionally, human studies that simply measure a proxy for usefulness (i.e. measuring human judgment of explanations) are often uninformative, as they may not translate into improvements in practice (Adebayo et al., 2018).”
>     - “LLM interpretations rarely come with theoretical guarantees of reliability, and those that do are often for very constrained settings.”

---

### Review · Reviewer_57Ff · 2024-06-24

**Summary Of Contributions:**

This paper reviews existing methods to evaluate the emerging field of LLM interpretation. Authors contend that LLMs hold the opportunity to redefine interpretability. Besides, authors highlight two emerging research priorities for LLM interpretation.

**Audience:**

Yes

**Claims And Evidence:**

Yes

**Requested Changes:**

(1) For Section 2, the definition of interpretability is now defined through text. If this definition can be defined mathematically, it will be more rigorous. Moreover, authors do not introduce existing evaluation methods in a clear order. I suggest that authors should first introduce there exist how many types of evaluation methods, and then detailedly introduce the goal, the advantage, and the disadvantages of each type of evaluation methods.

(2) Some claims in Section 3 authors do not explain clearly. For example, why “natural language allows for expressing complex concepts through explanations at different levels of granularity?” What are “different levels of granularity” are referred to? Can authors provide some examples to detailedly explain this claim?

(3) I think the first challenge of LLM interpretation is inappropriate, because “hallucination” is one kind of the “incorrect or baseless explanations”, rather than “hallucination that is incorrect or baseless explanations.” Moreover, authors do not mention/consider the LLM interpretation faces the challenge of leaking privacy, unfairness, etc. Hence, I think authors should conclude the challenge of LLM interpretation.

(4) The novelty of this paper is in doubt. For example, in Section 4, authors just list existing methods for explaining methods or citing others’ claims without proposing authors’ own thoughts or concluding the advantage/disadvantage/risk of these methods. Simply listing previous methods does not bringing in novelty.

(5) Minor: authors state that the focus of this paper is “the unique opportunities and challenges presented by LLMs,” but authors use very little space to introduce the opportunities and challenges. I think authors should spend more space discussing opportunities and challenges, rather than simply listing existing methods.

**Strengths And Weaknesses:**

This paper focuses on the LLM interpretation, which is a significantly important topic in XAI.

---

> ### Author Response · Authors · 2024-07-08
> **Thank you for your review**
>
> We sincerely thank the reviewer for their thoughtful comments and time. We have made several changes to the paper in the revised version to incorporate the feedback, resulting in a much improved paper.
>
> **(1) Defining interpretability** – We appreciate the desire for mathematically defining interpretability, but do not believe we can do so in a way that is clear and reasonable. As we note in the paper, “Precisely defining interpretability requires understanding the problem and audience an interpretation is intended to serve. In light of this imprecision, interpretable ML has largely become associated with a narrow set of techniques, including feature attribution, saliency maps, and transparent models. However, LLM interpretation is broader in scope and more expressive than these methods.” Efforts to mathematically define interpretability have led them to quickly diverge from the context in which they are used, even sometimes failing basic sanity checks [1]. For these reasons, we believe we cannot mathematically define interpretability in a way that is meaningful beyond textual definition we give.
>
> Re: Evaluation methods – We move evaluation into its own section (2.2) and add some new text to help make the section clearer and more thorough, e.g. “Evaluating interpretability is difficult in general (Doshi-Velez & Kim, 2017), requiring that interpretations be faithful to the underlying process they describe while remaining understandable to a human. This process is made even more difficult by the use of LLMs, which broadens the space of explanations and complicates rigorous understanding. In this section, we briefly overview three key approaches to evaluating LLM interpretations through (i) human studies, (ii) automated metrics, and (iii) using the interpretation to improve model performance. Careful consideration is required to match the choice of metric to the context of a particular problem.” We then clarify the text in each of these three categories as suggested.
>
> **(2) Examples** – We edit some of the text in Sec 3 to clarify some of the terse writing. For example, we add this note: “For example, a natural language explanation for an LLM's answer to a question may highlight the LLM's coarse reasoning if the LLM is prompted for a high-level explanation or instead to specific words in the input if the LLM is asked for a more fine-grained explanation.”
>
> **(3) Hallucination** – We clarify that we focus on hallucinated explanations rather than hallucinations in general, adding the following note:  “First and foremost is the issue of *hallucination*, i.e. incorrect or baseless LLM generations. In our setting, we focus on hallucinated explanations generated by an LLM.”
>
> We add the following paragraph to the end of section 2.1 describing interpertability’s relationship to these other areas in the context of LLMs: “Interpretability is intricately related to other research areas where LLMs have begun to play an expanding role. For example, in the field of causal inference, interpreting and querying LLMs may help extract and test hypotheses for causal relationships contained in data (Kıcıman et al., 2023). Additionally, interpretability is a major topic when considering the bias, fairness, privacy, and security of LLMs (Yao et al., 2024; Li et al., 2023d), where it can be used to expose, fix, or exploit issues with an LLM. Evaluating LLM interpretations We briefly overview three main approaches to evaluating LLM interpretations that assess an interpretation’s through (i) human studies, (ii) automated metrics, and (iii) using the interpretation to improve model performance.
>
> **(4) Novelty** – Indeed, assessing novelty for a survey paper such as this can be difficult. We note that the bulk of the survey piece of the work is in sections 4 and 5 and that there does not seem to exist a strong survey paper covering these topics in the literature already (particularly for section on 5 on explaining a dataset with LLMs). Sections 1-3 and sections 6-7 contextualize, synthesize, and make new claims about different areas of LLM interpretability.
>
> **(5) Space on unique opportunities** – The paper does spend a fair bit of space (most of section 4 and section 5) on covering recent works. We view this as a core contribution of this paper: LLM interpretability is a rapidly growing field (90 of the papers we reference are published just since the start of last year) and contextualizing these is an important task. Moreover, these works serve as grounding for the emerging opportunities and challenges we discuss in the paper (especially in sections 1, 3, and 6).
>
> 89 of the papers we reference are published either this or last year (2024 or 2023), evidencing the fact that we are not aiming to give a comprehensive, historical account of interpretability, but rather a forward-looking research agenda through covering nascent topics such as self-verification, RAG, grounding, and self-improvement.
>
> [1] https://arxiv.org/abs/1810.03292

---

### Decision · Action_Editor_BB3t · 2024-08-05

**Recommendation:** Reject

**Comment:**

All three reviewers reached a consensus of rejecting the paper. Clear summarization of insights is suggested.

**Audience:**

All experts in deep learning will be interested in explainable AI, but the audience may expect more insightful analysis.

**Claims And Evidence:**

This is a position paper for interpretability of the LLM, which focuses on an important issue. However, some claims are not clearly introduced and lacks solid mathematical definition.